# Ellagitannins and Flavano-Ellagitannins: Red Wines Tendency in Different Areas, Barrel Origin and Ageing Time in Barrel and Bottle

**DOI:** 10.3390/biom9080316

**Published:** 2019-07-29

**Authors:** Zuriñe Rasines-Perea, Rémi Jacquet, Michael Jourdes, Stéphane Quideau, Pierre-Louis Teissedre

**Affiliations:** 1Unité de Recherche Œnologie, EA 4577, USC 1366 INRA, ISVV, Université de Bordeaux, F33882 Villenave d’Ornon, France; 2Institut des Sciences Moléculaires (CNRS-UMR 5255), Institut Européen de Chimie et Biologie, Université de Bordeaux, 2 rue Robert Escarpit, 33607 Pessac CEDEX, France

**Keywords:** *C*-glucosidic ellagitannins, flavano-ellagitannins, HPLC-UV-MS, red wine

## Abstract

During maturation and ageing in oak barrels polyphenolic compounds from oak wood, and particularly *C*-glucosidic ellagitannins, can be released from wood to the wine. These ellagitannins can be involved in oxidation reactions, affecting the wine’s organoleptic properties such as astringency. In this study *C*-glucosidic ellagitannins and flavano-ellagitannins, acutissimins A and B and epiacutissimins A and B, as well as mongolicain A, which is the result compound of acutissimin A oxidation, were identified and quantified. The quantification was carried out by HPLC-UV-MS in 185 commercial samples from different cultivar areas (Bordeaux and Rioja), different barrel oak wood (French oak barrels and American oak barrels) and different ageing periods. The results show differences between the two zones in terms of compound concentrations. Moreover, the ageing process in bottle for Bordeaux wines are unlike Rioja wines behavior in bottle.

## 1. Introduction

Winemakers have been using wooden barrels for centuries as a common practice for wine ageing. This process is a way to improve a wine’s aroma, colour and mouthfeel [1,2,3] due to the extraction of specific wood chemical compounds, such as volatile phenols, lignins and hydrolysable tannins into wine [4]. Ellagitannins together with the gallotannins belong to the hydrolyzable tannins family; this term refers to their ability to release ellagic acid under acidic conditions [5,6]. Today, over 500 members of these gallic acid-derived polyphenolic natural products have been isolated from various plants and fully characterized [5,7].

Castalagin and vescalagin were the first two ellagitannin isomers to be isolated and characterized [8]. They present a structural specificity of having a highly characteristic C-C linkage between the carbon-1 atom of an open-chain glucose core and the carbon-2’ atom of a galloyl-derived unit esterified to the two-position of the glucose core (Figure 1). Another biarylic unit of hexahydroxydiphenoyl (HHDP) is linked on the four-, and six-position of the glucose. Lyxose/xylose derivatives (grandinin and roburin E, respectively) and dimeric forms (roburins A, B, C, D) have also been described [9,10] (Figure 1).

Several factors as the oak species (*Quercus robur* L., *Quercus petraea Liebl*.), the geographical origin [11,12,13], the age [14], the topography [15,16] and the geographic location of the wood piece in the trees [14,17] as well as the processing of wood in cooperage such as the type and duration of seasoning and toasting [18,19,20,21] can influence the content of the ellagitannins in oak wood [9]. Moreover, the age of the barrels is also important, since it has been reported that the levels of extracted ellagitannins are much lower in old barrels [22,23].

From the first contact of wine with the oak barrel, the *C*-glucosidic ellagitannins are slowly but continuously transformed through condensation, hydrolysis, and oxidation reactions. In the last years, the formation of flavano-ellagitannins (**9**–**12**) and the β-1-*O*-ethylvescalagin (**13**) in red wines aged in oak barrels has been reported [6,24,25,26,27] (Figure 1). Moreover, the acutissimin A is the precursor of mongolicain A (**14**) [28] (Figure 1), which was isolated for the first time inside *Quercus mongolica* in 1988 [29]. These *C*-glucosidic ellagitannins and flavano-ellagitannin hybrids exhibit important biological properties such as antioxidant, antitumoral, anti-inflammatory, antibacterial, and antiviral [6,30]. They can also affect the astringency of the wine, since, as procyanidins, they have the ability to precipitate proteins, in particular the salivary proteins in the oral cavity [1,2,31,32]. Finally, ellagic tannins can take part in the colour stability during maturation and ageing of wine and also protecting it against oxidation [1,33,34].

Some studies about red wine aged in oak barrels have been performed [33,35,36]. Recently a study of different cork stoppers during the ageing period in the bottle of wine model solution has demonstrated the implication of the type of cork on the concentration of *C*-glucosidic ellagitannins and flavano-ellagitannins, as well as for mongolicain [37]. However, all those studies have focused almost exclusively on one wine-producing area and with an ageing time between 0 to 12 months. Thus, the aim of this study was to monitor and compare the kinetics evolutions of the *C*-glucosidic ellagitannins **1**–**8**, the flavano-ellagitannins **9**–**12**, the β-1-*O*-ethylvescalagin (**13**) and the oxidation product of acutissimin A, mongolicain A (**14**) between commercial red wines from Bordeaux region (France) aged in French oak barrels and with several bottle ageing period and commercial red wines from Rioja region (Spain) aged in American oak barrels and also with different time periods in barrel and in bottle.

## 2. Materials and Methods

### 2.1. General

Chlorogenic acid (>95%), formic acid (>95%) and acetic acid were purchased from Sigma-Aldrich (St Quentin Fallavier, France). Methanol (HPLC grade quality, >99.8%) and TSK gel HW 50F were purchased from VWR (Strasbourg, France). Acetone was purchased from Xilab (Atlantic labo, Bordeaux, France). Milli-Q (Millipore) water was prepared using a Sarterius-arium 611 system.

Castalagin was extracted from Quercus robur heartwood [27] and the hemisynthesis of the acutissimin A/B (9/10, >95% pure), the epiacutissimin A/B (11/12, >95% pure), and the β-1-*O*-ethylvescalagin (13, 98% pure) were performed in acidic media [1.5% (*v*/*v*) TFA/THF] at 60 °C using (-)-vescalagine as previously described [25,26,27,28]. Castalagine and vescalagine were identified on the basis of their retention time, molecular mass, and mass fragmentation data. The identity of the hemisynthetic compounds was confirmed by comparison of 1H and 13C NMR data as well as optical rotations with published data [24]. The oxidation of acutissimin A to achieve the compound called mongolicain A was performed in acid aqueous solution. 100 mg of acutissimin A were placed in 2 mL of water acidified a pH 3.5 with tartaric acid. This aqueous solution was behind air contact and environmental temperature (25 °C) for 15 days.

### 2.2. Red Wines

Eighty-five commercial red wine samples of the Bordeaux region from the wine-making areas of Medoc, Graves, Libournais, Blayais & Bourgeais as well as from the generic origin appellation of Bordeaux, that were aged for 12 months in French oak barrel and with different time in bottle (from one to 10 years; production years: from 1994 to 2010) were analyzed in this research. In addition, 100 samples from the three wine-producing areas of Rioja (Rioja Alta, Rioja Baja and Rioja Alavesa), three different ageing period in American oak barrels (12, 24 and 36 months) and also with different times in bottle (from two to 15 years; production years: from 2000 to 2013) were also analyzed. As all the analyzed samples are commercial red wine bottles, further information regarding the ageing process and oak barrel conditions are not reported on bottle label.

### 2.3. Sample Preparation

A fractionation procedure used prior to HPLC-UV-MS analysis of the red wine samples was adapted from the previously described procedure [27,35]. Briefly, 120 mL of each red wine sample was evaporated under vacuum and dissolved in H_2_O/HCOOH (996:4, 20 mL). Solutions were then loaded on a column (55 mm × 25 mm) that had been packed with TSK HW 50F resin, which had been previously swelled overnight in methanol and equilibrated with H_2_O/HCOOH (996:4, 100 mL). After loading the sample on the column, the acidic aqueous solvent (50 mL) was first used to wash out tartaric acid and sugars and, in a second step, an acidic hydromethanolic solvent (H_2_O/MeOH/HCOOH (298:698:4), 100 mL) was used to elute important amounts of the non-ellagic polyphenols.

Finally, the ellagitannin fraction was eluted using H_2_O/acetone/HCOOH (298:698:4, 100 mL). This fraction was evaporated under reduced pressure to furnish a reddish light brown residue, which was dissolved in H_2_O/HCOOH (996:4, 1 mL) containing 20 mg/L of chlorogenic acid used as internal standard and filtered (0.45 µm) prior to HPLC-UV-MS analysis.

### 2.4. HPLC-UV-MS Conditions for Ellagitannins and Flavano-Ellagitannins Quantification in Wine

The methodology for the analysis of ellagitannins composition in red wines was adapted from the previously described procedure [2]. The analysis of ellagitannins composition in red wines was performed using a Thermo-Finnigan Surveyor HPLC system containing an autosampler (Surveyor autosampler Plus), a quaternary pump (Surveyor LC pump Plus), and a UV-vis detector (Surveyor PDA Plus) controlled by Xcalibur data treatment system. This HPLC system was also coupled to a Thermo-Finnigan LCQ Advantage spectrometer equipped with an ion trap mass analyzer. The electrospray ionization and mass detection were performed in negative ion mode with the following optimized parameters: capillary temperature, 300 °C; capillary voltage, -46 V; nebulizer gas flow, 1 L/min; desolvation gas flow, 0.25 L/min; and spray voltage, 5 kV. These analyses were carried out in technical duplicates on a 250 × 4.6 mm, 5 µm, Lichrospher 100 RP 18 column. The mobile phases used were solvent A [H_2_O/HCOOH (996:4)] and solvent B [MeOH/HCOOH (996:4)] and gradient elution of 0–3% of B in 5 min, 3-2 20% of B in 20 min, and 20–100% of B in 50 min with a flow rate set at 1 mL/min and a detection wavelength set at 280 nm. The C-glucosidic ellagitannins (**1**–**8**), flavano-ellagitannins (**9**–**12**), mongolicain A (**13**) and β-1-*O*-ethylvescalagin (**14**) were separately identified and assigned by comparison of their UV spectra and mass spectra with that of purified or hemisynthesized standards. The quantification of each compound was performed using their molecular ion (Table 1). The concentrations of C-glucosidic ellagitannins and β-1-*O*-ethylvescalagin were expressed as equivalents of castalagin, whereas flavano-ellagitanins and mongolicain A concentrations were expressed as equivalents of acutissimin A.

## 3. Results and Discussion

### 3.1. Concentrations and Composition of Ellagitannins and Flavano-Ellagitannins Depending on the Wine-Producing Area

In order to evaluate exclusively the factors of region and wood barrel type, only the wines produced in the 2010 vintage with the same ageing time (1 year) are presented in this section.

Regarding the vineyard zones from Bordeaux, there are important differences between wine regions for the total concentrations on ellagitannins (Figure 2a), with a maximum of 7.98 mg/L for the wines of Libourne and a minimum of 3.99 mg/L for Bordeaux wine generic origin appellation. For all the vineyards, the ellagitannin with the highest concentration is castalagin followed by mongolicain A, with percentage values between 57% and 70% for castalagin and between 16% and 27% for the mongolicain A. It is remarkable that in Blaye & Bourges wines acutissimins or epiacutissimins were not detected, while for generic wines no β-1-*O*-ethyl-vescalagin was found.

Making the same comparison for Rioja wines (Figure 2b), there is not such difference between the 3 zones when it comes to ANOVA results for the total sum of ellagitanins. Moreover, the percentages of main compounds were practically the same with values ranging from 52 to 56% for castalagin, 8–11% for Roburin E, 5–9% for Mongolicain A and 7–9% for B-1-*O*-Ethylvescalagin.

Comparing all the results together, the concentration average for the sum on ellagitannins of Bordeaux wine samples reached 5.582 mg/L, two times higher than the concentration determined for Rioja wines.

In terms of percentages, castalagin shows the greatest concentration with participation of 64 and 57% for samples from Bordeaux and Rioja, respectively. After castalagin the mongolicain A is the second molecule present in majority for Bordeaux wines with a contribution of 18.46%, three-folds higher than the value obtained for Rioja wines. This difference can be explained by the participation of vescalagin, ethyl-vescalagin, roburin E and epiacutissimin B in Rioja wines with percentages of 3.86%, 9.18%, 11.27% and 5.94% respectively. Whereas the majority of these compounds have, values lower than 2% in Bordeaux wine samples.

### 3.2. Concentrations and Composition of Ellagitannins and Flavano-Ellagitaninins Depending on the Ageing Time in Oak Barrels for Rioja Wines

Looking to the concentration on ellagitannins depending on the barrel ageing time from Rioja wines (Figure 3), the total content for the wines that have been kept for 24 months (Reserva) in oak barrels was the half of the ellagitannin content in wines that have remained only 12 months (Crianza) in barrels. This coincides with other research [35] where it is visible a decrease of total concentration on ellagitannins through the time. But when we look at the wines with 36 months (Gran Reserva) of ageing in oak barrels their concentration is remarkable, which reaches 2.76 mg/L. This result can be explained by the large options of barrels types, in terms of grain, porosity, toasting and drying conditions and wood origin and species [38] that are found inside barrel-making industry and other unknown factors such as the age of the barrels and the number of times that the barrels have been used and also their sanitization status inside wineries.

### 3.3. Concentrations and Composition of Ellagitannins and Flavano-Ellagitaninins Depending on the Ageing Time in Bottle

Figure 4 shows the content of each ellagitannin and flavano-ellagitannin in wines during the ageing process from Libourne, Blaye & Bourges and Rioja wine producing-area, respectively. As it is visible, the contents in ellagitannins and flavano-ellagitannins for Liboune and Blaye & Bourges wines (Figure 4a,b) decrease during the time in bottle, even the concentration of castalagin, which is more stable than vescalagin, is lower in more aged wines. What is more, these results suggest that not only the ellagitannins but also the flavano-ellagitannins are degraded during the ageing period, as it is visible the lack of flavano-ellagitannins and mongolicain A in the most aged wines. However, Rioja wines (Figure 4c) have an unlike behavior from Libourne and Blaye & Bourges wines, where it is visible a stability during the years, which can be explained by the procedure that allows blending the final wine with a maximum of 15% of another wine. According to the Rioja’s origin appellation regulation, this wine percentage should come from the same vintage but could have different oak barrel conditions (old or new barrel, French or American barrel), which are unknown. This practice could increment the content on ellagitannins that can end in the stabilization of the wine inside the bottle.

## 4. Conclusions

Our study shows that the evolution of the *C*-glucosidic ellagitannins concentration during red wine ageing depends on the type of oak barrels, the time stayed inside the barrels and the period in the bottle. For the red wine aged in French oak barrels, the *C*-glucosidic ellagitannins average concentration in Bordeaux wines was two-folds higher than the value obtained for Rioja wines aged in American oak barrels. These differences in concentration levels could have an impact on the wine gustative properties such as astringency, bitterness, amplitude, and roundness of the wine.

The *C*-glucosidic ellagitannins and flavano-ellagitannins concentrations depending on the three different origin appellations qualification of Rioja, “*Crianza*”, “*Reserva*” and “*Gran Reserva*”, that involve times in oak barrels from 12 to 36 months and periods in bottle from 12 to 36 months before their consumption, were analyzed for the first time. Moreover, the practice in Rioja region for the origin appellation that allows the addition of 15% of another type of wine from the same vintage could provide more stable concentration in ellagitannins and flavano-ellagitannis.

In this comparative study between Bordeaux and Rioja wines, the concentration on mongolicain A in Bordeaux wines is three times higher than for Rioja wines, suggesting that this compound could be a marker of the time of ageing in oak barrel. The degradation of *C*-ellagitannins and flavano-ellagitannins is more accused in Rioja wines, which were kept longer in oak barrels and showed the lowest concentrations on this type of compounds.

Further investigation should be performed in order to estimate the overall impact of these *C*-glucosidic ellagitannins and vescalagin derivatives on red wine organoleptic perception.

## Figures and Tables

**Figure 1 biomolecules-09-00316-f001:**
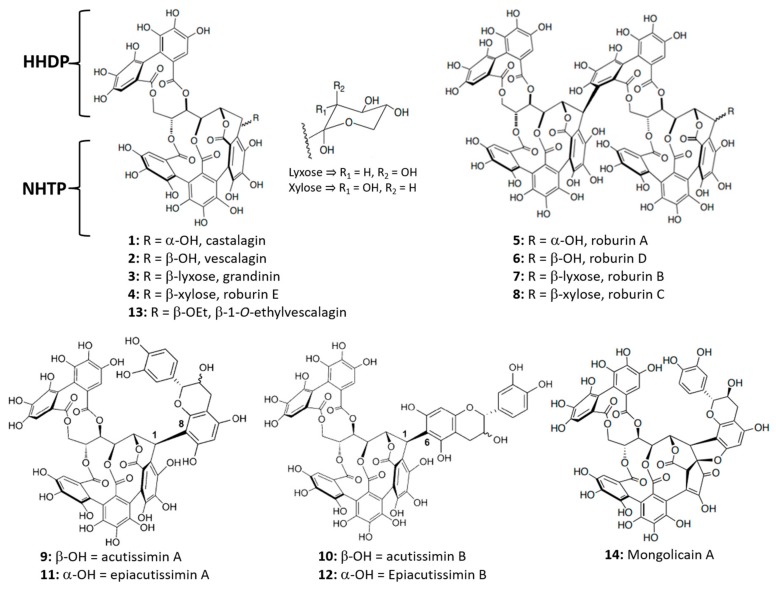
Structures of *C*-glucosidic ellagitannins **1**–**8** found in heartwood of Quercus species, structures of the vescalagin derivative **9**–**13** and oxidation product **14**, identified and quantified in red wine aged in contact with oakwood.

**Figure 2 biomolecules-09-00316-f002:**
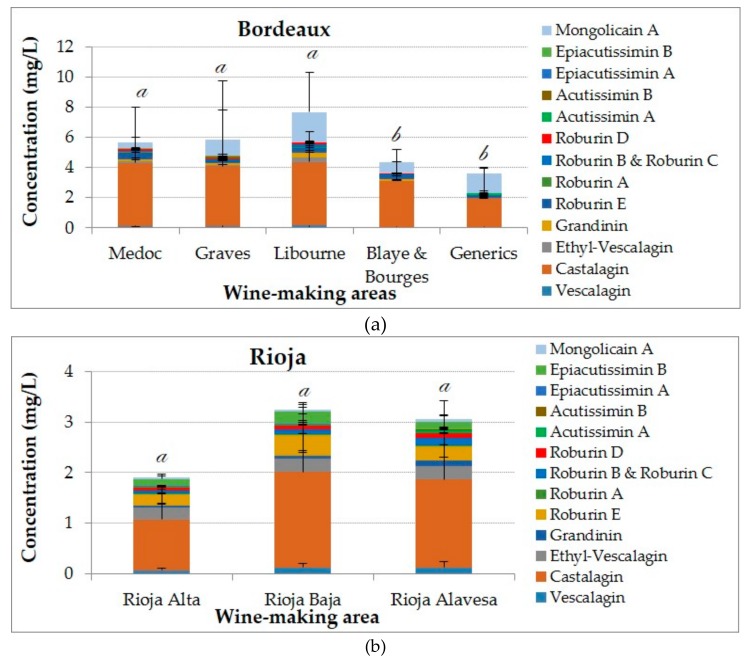
Concentrations and composition of ellagitannins and flavano-ellagitannins depending on the wine-producing area: (**a**) Bordeaux zones; (**b**) Rioja zones. The ANOVA tests performed compare the values between regions for a given analysis. The different letters indicate a significant difference between the values achieved for total ellagitannins (Tukey’s test, *p* < 0.05).

**Figure 3 biomolecules-09-00316-f003:**
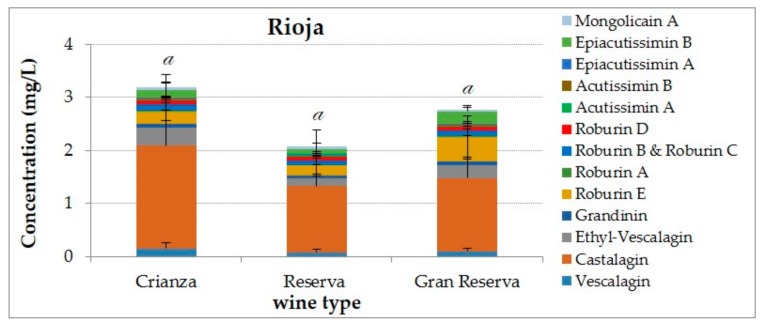
Concentrations and composition of ellagitannins and flavano-ellagitannins depending on the ageing time in oak barrels for Rioja wines. The ANOVA tests performed compare the values between categories for a given analysis. The different letters indicate a significant difference between the values achieved for total ellagitannins (Tukey’s test, *p* < 0.05).

**Figure 4 biomolecules-09-00316-f004:**
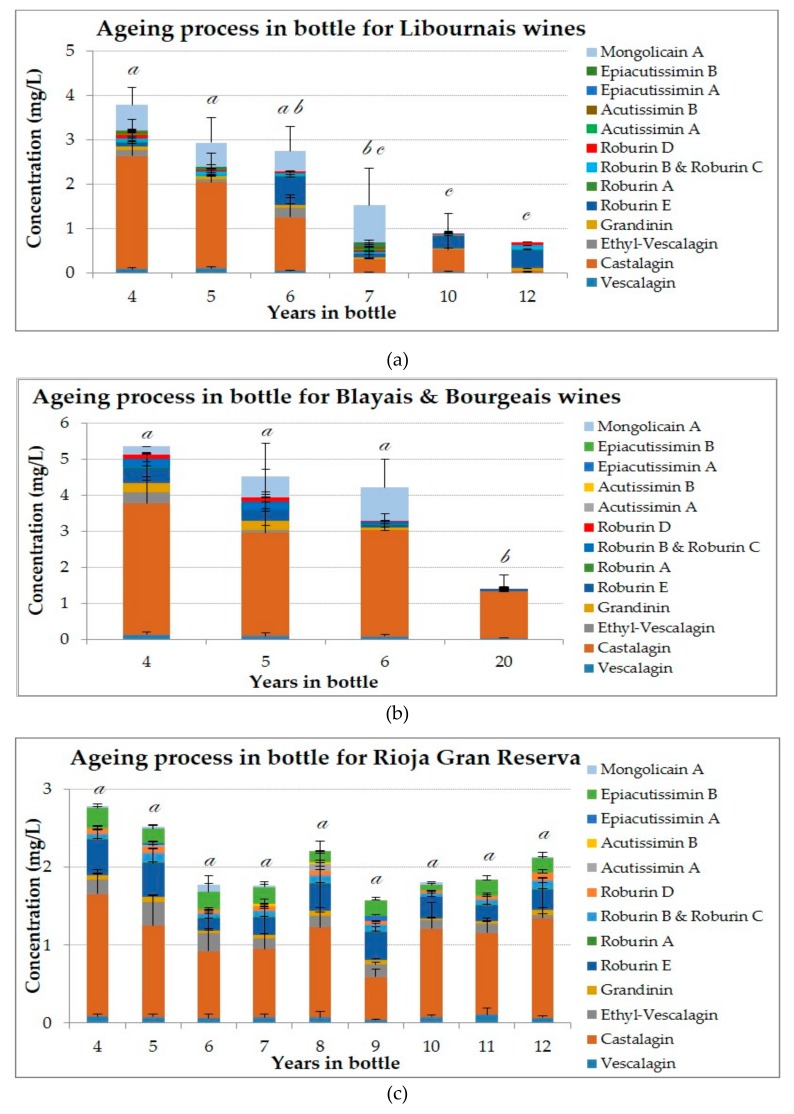
Concentrations and composition of ellagitannins and flavano-ellagitannins depending on the ageing period in bottle: (**a**) Libourne wines; (**b**) Blaye & Bourges wines; (**c**) Rioja wines. The ANOVA tests performed compare the values between years in bottle for a given analysis. The different letters indicate a significant difference between the values achieved for total ellagitannins (Tukey’s test, *p* < 0.05).

**Table 1 biomolecules-09-00316-t001:** HPLC retention times and mass fragmentation patterns of the reference compounds **1**–**14** and the internal standard.

Compounds	Retention Time (min)	m/z
**Castalagin (1)**	22.80	933 ^a^, 915, 613, 301
**Vescalagin (2)**	13.03	933 ^a^, 915, 613, 301
**Grandinin (3)**	8.21	1065 ^a^, 915, 613, 301
**Roburin A (5)**	7.42	1849, 933, 924 ^a^, 915, 301
**Roburin B (7)**	7.92	1981, 1065, 990 ^a^, 915, 301
**Roburin C (8)**	7.96	1981, 1065, 990 ^a^, 915, 301
**Roburin D (6)**	10.55	1849, 933, 924 ^a^, 915, 301
**Roburin E (4)**	15.16	1065 ^a^, 915, 613, 301
**Acutissimin A (9)**	50.62	1205 ^a^, 915, 613, 602, 301
**Acutissimin B (10)**	62.02	1205 ^a^, 915, 613, 602, 301
**Epiacutissimin A (11)**	67.49	1205 ^a^, 1053, 915, 613, 602, 301
**Epiacutissimin B (12)**	44.92	1205 ^a^, 1053, 915, 613, 602, 301
**β-1-*O*-ethylvescalagin (13)**	42.32	961 ^a^, 915, 480, 301
**Mongolicain A (14)**	54.50	1175 ^a^
**Chlorogenic acid**	56.63	353 ^a^

a: Ion used for the quantification.

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
