# Peer review of "Ellagitannins and Flavano-Ellagitannins: Red Wines Tendency in Different Areas, Barrel Origin and Ageing Time in Barrel and Bottle"

_biomolecules, 2019, doi:10.3390/biom9080316_

Round 1
Reviewer 1 Report
The manuscript cannot be accepted in his present form but it needs major revision. In particular more information related to aged red wines and the kind of barrels used should be added in materials and methods section. Do the authors have information on the 15% wines added in Rioja?
All these information could help discussion on results.
The English should be revised (e.g. lines 27, 43…).
Paragraph 2.2. If possible, please add information on red wine aged conditions and year of production.
Paragraph 2.3. Please briefly describe the fractionation procedure.
Lines 136-139. Without the knowledge of the kind of wood used for aging and how long wines were aged in barrels this results cannot be related to differences in aging procedures used in Bordeaux and Rioja.
Lines 164-166. This sentence is not clear.
Author Response
You will find there our answers:
The manuscript cannot be accepted in his present form but it needs major revision. In particular more information related to aged red wines and the kind of barrels used should be added in materials and methods section. Do the authors have information on the 15% wines added in Rioja?
All these information could help discussion on results.
As it is described in paragraph 2.2, samples are commercial wine bottles. So the aging process in oak barrels and barrels conditions are, mostly not specified in bottle etiquette. We have made an indication by including the sentences of “As all samples are commercial red wine bottles, information of aging process and oak barres conditions are not repported on bottle etiquette.” in paragraph 2.2.
The English should be revised (e.g. lines 27, 43…).
Paragraph 2.2. If possible, please add information on red wine aged conditions and year of production.
As suggested by the reviewer, Paragraph 2.2 has been modified including the following sentences:
… (from 1 to 10 years; production years: from 1994 to 2010)…: For Bordeaux wines
… (from 2 to 15 years; production years: from 2000 to 2013)…: For Rioja wines
“As all samples are commercial red wine bottles, information of aging process and oak barres conditions are not repported on bottle etiquette.” At the end of paragraph 2.2.
Paragraph 2.3. Please briefly describe the fractionation procedure.
As suggested by the reviewer, Paragraph 2.3 has been modified including the following sentence:
“Where 120mL of each red wine sample was evaporated under vacuum and dissolved in H2O/HCOOH (996:4, 20 mL). Solutions were then loaded on a column (55 mm × 25 mm) that had been packed with TSK HW 50F resin, which had been previously swelled overnight in methanol and equilibrated with H2O/HCOOH (996:4, 100 mL). After loading the sample on the column, the acidic aqueous solvent (50 mL) was first used to wash out tartaric acid and sugars and, in a second step, an acidic hydromethanolic solvent (H2O/MeOH/HCOOH (298:698:4), 100 mL) was used to elute important amounts of the non ellagic polyphenols”.
Lines 136-139. Without the knowledge of the kind of wood used for aging and how long wines were aged in barrels this results cannot be related to differences in aging procedures used in Bordeaux and Rioja.
Answer: authors modified the paragraph as the graphics refers to bottles of vintage 2010 with same period in bottle and barrel. The variable is the type of wood barrel and region.
Lines 164-166. This sentence is not clear.
Answer: authors have modified the sentence. This 15 % refers to the oak barrel description for this percentage (new or old, French or American). Aging conditions could be different for this 15 % of wine and thus affect the final composition of ellagitannins in wine.
Reviewer 2 Report
The research has an interesting approach about the incidence of different factors in the presence of ellagitannins in aged red wines, but from my point of view a substantial change is necessary, since I notice important handicaps in both, the statistical analysis and the discussion.
The statistical analysis of obtained data is missing: nor univariate nor multivariate analysis were performed. It is not possible to know if the differences found in the levels of ellagitannins in the studied wines were statistically significant. Nor is it possible to know compounds that contribute most significantly to the differences between the analyzed wines in relation to the different factors studied (origin, wine-producing area, time in barrel, time in bottle).
The discussion of results is also almost lacking, particularly about the incidence of wood barrel origin (French vs American), but also about barrel ageing time in Rioja, or about bottle ageing time:
-Comments in Ln 148-151 are not enough and included some not realistic comments, such as those regarding “wood origin and species” (only American oak Q. alba is used), to differentiate between “the age of the barrels and the number of times that the barrels have been used”, or to take on “their sanitization status inside wineries” (Is their state of sanitation within the cellars related to the level of ellagitannins in the wines?).
-Comments in Ln 164-167 are also not enough, and a more deep discussion should be included. On the other hand, the claim that in Rioja 15% of another wine can be added is wrong (Ln 164-167 and Ln187-189). This practice is established only in relation to the determination of the vintage year, but in no case does it refer to the aging time in barrels or bottles that the wines must accomplish in order to obtain the crianza, reserva or gran reserva qualification.
Other concerns:
-Ln 13, 16, 28, 29, 36, and throughout the document: check spelling, spaces, and punctuation marks. Use italics for the species.
-Ln 40: only two Quercus species are used in cooperage??
-Ln 43: replace drying by seasoning
-Figure 1. Structures 9-12 - I have not found R in the chemical structure drawn
-Both, the sample preparation and the methodology for ellagitannins analysis, were adapted from previously described procedures, but the authors do not explain how they carried out the adaptation. It should be included a more detailed description.
-Table 1: I think it would be better not to include the comma in the m/z higher than 1000, such as for mongolicain.
-Figures 2-4: Roburin B appear twice in the legends. The typeface should always be the same. Are concentrations the average values? What do the error bars represent: standard deviation or standard error?
Section 3.3. Why is only a part of the data included? What happens in Medoc, Graves, Generic Bordeaux, crianza, reserva?
-Figure 4: the year on the abscissa, does it refer to the year of harvest? I think that in this axis should appear ageing years in bottle (from 1 to 10, or from 2 to 15).
-Ln181: add “aged in American oak barrels” after Rioja wines.
Ln 184-185: Crianza, Reserva and Gran Reserva: they are not 3 different origin appellations of Rioja but 3 qualifications in relation to ageing times in barrel and in bottle
Author Response
You will find there our answers:
Comments and Suggestions for Authors
The research has an interesting approach about the incidence of different factors in the presence of ellagitannins in aged red wines, but from my point of view a substantial change is necessary, since I notice important handicaps in both, the statistical analysis and the discussion.
The statistical analysis of obtained data is missing: nor univariate nor multivariate analysis were performed. It is not possible to know if the differences found in the levels of ellagitannins in the studied wines were statistically significant. Nor is it possible to know compounds that contribute most significantly to the differences between the analyzed wines in relation to the different factors studied (origin, wine-producing area, time in barrel, time in bottle).
The discussion of results is also almost lacking, particularly about the incidence of wood barrel origin (French vs American), but also about barrel ageing time in Rioja, or about bottle ageing time:
-Comments in Ln 148-151 are not enough and included some not realistic comments, such as those regarding “wood origin and species” (only American oak Q. alba is used), to differentiate between “the age of the barrels and the number of times that the barrels have been used”, or to take on “their sanitization status inside wineries” (Is their state of sanitation within the cellars related to the level of ellagitannins in the wines?).
Answer: This article is a first approach about the different tendencies between commercial red wine of different regions, different barrel types, different time in barrel and time in bottle. On the other hand, when we discuss of “wood origin and species”, “the age of the barrels and the number of times that the barrels have been used”. We only have the bottle etiquette information and is well known that sometimes a little percentage of French oak is used also in the aging of Rioja wines (that can influence to the difference). Moreover, the sentence referring “the age of the barrels and the number of times that the barrels have been used” may be is not clear but it is correct, reference [38] explain the difference between new and old barrels in terms of ellagitannins concentration.
The sentence of “their sanitization status inside wineries” has been eliminated.
-Comments in Ln 164-167 are also not enough, and a more deep discussion should be included. On the other hand, the claim that in Rioja 15% of another wine can be added is wrong (Ln 164-167 and Ln187-189). This practice is established only in relation to the determination of the vintage year, but in no case does it refer to the aging time in barrels or bottles that the wines must accomplish in order to obtain the crianza, reserva or gran reserva qualification.
Answer: authors have modified the sentence. This 15 % refers to the oak barrel description for this percentage (new or old, French or American). Aging conditions could be different for this 15 % of wine and thus affect the final composition of ellagitannins in wine.
Other concerns:
-Ln 13, 16, 28, 29, 36, and throughout the document: check spelling, spaces, and punctuation marks. Use italics for the species.
-Ln 40: only two Quercus species are used in cooperage??
Authors give only an example of Quercus species. Ln 40 is part of the introduction, not of materials and methods.
-Ln 43: replace drying by seasoning done
-Figure 1. Structures 9-12 - I have not found R in the chemical structure drawn
As reviewer suggest, figure 1 was modified for structures 9-12.
-Both, the sample preparation and the methodology for ellagitannins analysis, were adapted from previously described procedures, but the authors do not explain how they carried out the adaptation. It should be included a more detailed description.
As reviewer suggested the following paragraphs have been included:
-For sample preparation:
“Where 120mL of each red wine sample was evaporated under vacuum and dissolved in H2O/HCOOH (996:4, 20 mL). Solutions were then loaded on a column (55 mm × 25 mm) that had been packed with TSK HW 50F resin, which had been previously swelled overnight in methanol and equilibrated with H2O/HCOOH (996:4, 100 mL). After loading the sample on the column, the acidic aqueous solvent (50 mL) was first used to wash out tartaric acid and sugars and, in a second step, an acidic hydromethanolic solvent (H2O/MeOH/HCOOH (298:698:4), 100 mL) was used to elute important amounts of the non ellagic polyphenols”.
-For analyse:
“The analysis of ellagitannins composition in red wines was performed using a Thermo-Finnigan Surveyor HPLC system containing an autosampler (Surveyor autosampler Plus), a quaternary pump (Surveyor LC pump Plus), and a UV-vis detector (Surveyor PDA Plus) controlled by Xcalibur data treatment system. This HPLC system was also coupled to a Thermo-Finnigan LCQ Advantage spectrometer equipped with an ion trap mass analyzer. The electrospray ionization and mass detection was performed in negative ion mode with the following optimized parameters: capillary temperature, 300 °C; capillary voltage, -46 V; nebulizer gas flow, 1 L/min; desolvation gas flow, 0.25 L/min; and spray voltage, 5 kV. These analyses were carried out in technical duplicates on a 250 × 4.6 mm, 5 µm, Lichrospher 100 RP 18 column. The mobile phases used were solvent A [H2O/HCOOH (996:4)] and solvent B [MeOH/HCOOH (996:4)] and gradient elution of 0-3% of B in 5 min, 3-2 20% of B in 20 min, and 20-100% of B in 50 min with a flow rate set at 1 mL/min and a detection wavelength set at 280 nm”
-Table 1: I think it would be better not to include the comma in the m/z higher than 1000, such as for mongolicain.
The authors have eliminated the commas appering in table 1.
-Figures 2-4: Roburin B appear twice in the legends. The typeface should always be the same. Are concentrations the average values? What do the error bars represent: standard deviation or standard error?
The error bars represent standard deviation.
Graphics legends have been corrected.
-Section 3.3. Why is only a part of the data included? What happens in Medoc, Graves, Generic Bordeaux, crianza, reserva?
From the total of 185 samples collected the authors have decided to show only the graphics for the regions or wine type with a high number of different production years. For Medoc, Graves, Generic Bordeaux, Crianza and reserva there were not enough production year to have a clear tendency.
-Figure 4: the year on the abscissa, does it refer to the year of harvest? I think that in this axis should appear ageing years in bottle (from 1 to 10, or from 2 to 15).
As reviewer has suggested, authors have modified graphics absissa.
-Ln181: add “aged in American oak barrels” after Rioja wines. done
-Ln 184-185: Crianza, Reserva and Gran Reserva: they are not 3 different origin appellations of Rioja but 3 qualifications in relation to ageing times in barrel and in bottle
The sentence has been modified.
Reviewer 3 Report
This work is of particular interest and provides new data on the evolution of ellagitannins and flavanol-ellagitannins during the maturation and ageing processes of wine in oak barrels obtained from different cultivar areas, different barrel oak wood origins as well as different ageing period. This work could also have important consequences in human health since some of these compounds are well described for their biological activities in human (the Authors could also point this aspect in their conclusion).
The method is well described. The figures are understandable even if it’s sometime difficult to follow because of the error bars overlay. The text is well written, provide sufficient background to follow the progression and justification of the present work in regards with the data available in the literature. The conclusions are supported by the results and therefore solid.
I recommend this work for publication in Biomolecules following the minor changes:
- In its present form the title did not take into account the maturation and ageing processes of wine in oak barrels. Please edit the title in order to include the main results of this work.
- There are misspellings and typing errors that are to be corrected. Please carefully revise the manuscript. Some examples given:
o L19 areas
o Choose between « La Rioja », « Rioja » or « la Rioja » that are used in the text for the same area
o Choose between “aging” and ageing”
o L135 mongolicain A
o L137 ethyl-vescalagin
o L144 months
o L148 barrel types
o L159 contents in
o L160 decrease
o L162 not only
Author Response
Comments and Suggestions for Authors
This work is of particular interest and provides new data on the evolution of ellagitannins and flavanol-ellagitannins during the maturation and ageing processes of wine in oak barrels obtained from different cultivar areas, different barrel oak wood origins as well as different ageing period. This work could also have important consequences in human health since some of these compounds are well described for their biological activities in human (the Authors could also point this aspect in their conclusion).
The method is well described. The figures are understandable even if it’s sometime difficult to follow because of the error bars overlay. The text is well written, provide sufficient background to follow the progression and justification of the present work in regards with the data available in the literature. The conclusions are supported by the results and therefore solid.
I recommend this work for publication in Biomolecules following the minor changes:
- In its present form the title did not take into account the maturation and ageing processes of wine in oak barrels. Please edit the title in order to include the main results of this work.
Authors change the article title for “Ellagitannins and flavano-ellagitannins: Red wines tendency in different areas, barrel origin and ageing time in barrel and bottle”
- There are misspellings and typing errors that are to be corrected. Please carefully revise the manuscript. Some examples given:
o L19 areas Donne
o Choose between « La Rioja », « Rioja » or « la Rioja » that are used in the text for the same area Donne
o Choose between “aging” and ageing” Donne
o L135 mongolicain A Donne
o L137 ethyl-vescalagin Donne
o L144 months Donne
o L148 barrel types Donne
o L159 contents in Donne
o L160 decrease Donne
o L162 not only Donne
Reviewer 4 Report
The manuscript describes the quantification of C-glucosidic ellagitannins and flavano-ellagitannins, acutissimins A 16 and B and epiacutissimins A and B, as well as mongolicain A in 185 samples of wines from different regions, barrel wood type and aging time. The topic is interesting and represents original results, but unfortunately the results are poorly presented and discussed, and material and methods were not adequately described. Major English editing is also required. My recommendation is to reconsider the manuscript after a major revision.
Suggestions:
1. Abstract: The abstract is poorly presented and very little informative of the main results obtained in the study. It should include the main results and a short conclusion of the work as a whole.
2. Species names (eg. Line 40, line 77) should be in italics.
3. Material and Methods: the authors report that castalagin was extracted from Quercus robus Heartwood and implied this compound as the starting material for the hemisynthesis of acutissimin A/B, epiacutissimin A/B, and ß-1-O-78 ethylvescalagin. However, in the paper by Quideau et al. (Chem. Eur. J. 2005, 11, 6503–6513), all nucleophilic reactions to obtain acutissimins A and B as well as epiacutissimins A/B were done with (-)-vescalagin as the strating material. What reactions conditions were used and what starting material was actually used for the hemisynthesis of the compounds cited in the manuscript?
4. The reaction conditions for the oxidation of acutissimin A into mongolicain A should be given.
5. The purity of the obtained compounds is given. What method was used to ascertain the purity of the compounds? The percentage purity given is HPLC peak purity? Is there any high-resolution mass spectral data to support the purity of compounds? This should be provided. A condensed form of the NMR spectral data should be provided to confirm identity of the substances.
6. The authors report that the quantification of compounds was done using a previously described LC/MS method (reference 2). However, no experimental details were given of their use of this method: For example: What were the separation conditions? What were the ionization conditions? Was a linear quadrupole ion trap mass spectrometer (Sciex, API4000 QTRap) used as in the original reference? What detection mode was used? The authors should provide typical ion chromatograms of the quantitated compounds in the wine fractions to prove that adequate selectivity was achieved.
7. The authors did not specify if the results presented in Figures 2 through 4 are mean values of replicates (n=?). Error bars are SD or SEM? Also, the statistical significance of the differences was ever considered? The way data was presented made it impossible to distinguish error bars for different compounds. This, associated with the lack of statistical analysis of the data makes the conclusions meaningless.
8. English is fairly poor and sloppy in some sentences, especially when presenting results. Some examples are given, but they are by no means a complete list and a thorough revision of English and grammar is mandatory. For example: Line 143: “…but without regarding the period stayed in bottle…” or line 130: “…Comparing all the results together, the concentration average for the sum on ellagitannins…” or line 157: “…Figures 4 shown the content of each ellagitannin…” or line 166: “…This practice could increment the content on ellagitannins that can end in the stabilization of the wine inside the bottle…” or lines 187 through 189: “…Moreover, the practice in Rioja region for the origin appellation that allows the addition of 15% of another type of wine, give concentration in ellagitannins ans flavano-188 ellagitannis more stable without depending on the oak barrel or bottle ageing periods.”
Author Response
There are our answers:
Comments and Suggestions for Authors
Suggestions:
1. Abstract: The abstract is poorly presented and very little informative of the main results obtained in the study. It should include the main results and a short conclusion of the work as a whole.
Abstract was modified by the authors.
...”After castalagin the mongolicain A is the second molecule present in majority for Bordeaux wines with a contribution of 18.46%, 3 folds higher than the value obtained for Rioja wines, suggesting that this compound could be a marker of the time of ageing in oak barrel. The degradation of C-ellagitannins and flavano-ellagitannins is more accused in Rioja wines, which were kept longer in oak barrels and showed the lowest concentrations on this type of compounds”
2. Species names (eg. Line 40, line 77) should be in italics. Done
3. Material and Methods: the authors report that castalagin was extracted from Quercus robus Heartwood and implied this compound as the starting material for the hemisynthesis of acutissimin A/B, epiacutissimin A/B, and ß-1-O-78 ethylvescalagin. However, in the paper by Quideau et al. (Chem. Eur. J. 2005, 11, 6503–6513), all nucleophilic reactions to obtain acutissimins A and B as well as epiacutissimins A/B were done with (-)-vescalagin as the strating material. What reactions conditions were used and what starting material was actually used for the hemisynthesis of the compounds cited in the manuscript?
Answer : We did not said that castalagine was used for flavano-ellagitannins hemysinthese. Castalagine was one of the ellagitannins analyzed in our work and to have the standard this compound was extracted from oak heartwood. It is (-)-vescalagine the compound used for the hemysintheses. The reactions conditions were the same as in references, because they were carried out by S. Quideau laboratory.
4. The reaction conditions for the oxidation of acutissimin A into mongolicain A should be given.
The oxidation of acutissimin A to achieve the compound called mongolicain A was performed in acid aqueous solution. 100 mg of acutissimin A were placed in 2 ml of water acidified a pH 3.5 with tartaric acid. This aqueous solution was behind air contact and environmental temperature (25°C) during 15 days.
5. The purity of the obtained compounds is given. What method was used to ascertain the purity of the compounds? The percentage purity given is HPLC peak purity? Is there any high-resolution mass spectral data to support the purity of compounds? This should be provided. A condensed form of the NMR spectral data should be provided to confirm identity of the substances.
All the hemysintheses were carried out inside S. Quideau laboratory with the same protocols as they were described on the references of the article. All the information reviewer is asking for is given in S.Quideau’s references.
6. The authors report that the quantification of compounds was done using a previously described LC/MS method (reference 2). However, no experimental details were given of their use of this method: For example: What were the separation conditions? What were the ionization conditions? Was a linear quadrupole ion trap mass spectrometer (Sciex, API4000 QTRap) used as in the original reference? What detection mode was used? The authors should provide typical ion chromatograms of the quantitated compounds in the wine fractions to prove that adequate selectivity was achieved.
As reviewer suggested, authors have added some missing details of quantifications :
The analysis of ellagitannins composition in red wines was performed using a Thermo-Finnigan Surveyor HPLC system containing an autosampler (Surveyor autosampler Plus), a quaternary pump (Surveyor LC pump Plus), and a UV-vis detector (Surveyor PDA Plus) controlled by Xcalibur data treatment system. This HPLC system was also coupled to a Thermo-Finnigan LCQ Advantage spectrometer equipped with an ion trap mass analyzer. The electrospray ionization and mass detection was performed in negative ion mode with the following optimized parameters: capillary temperature, 300 °C; capillary voltage, -46 V; nebulizer gas flow, 1 L/min; desolvation gas flow, 0.25 L/min; and spray voltage, 5 kV. These analyses were carried out in technical duplicates on a 250 × 4.6 mm, 5 µm, Lichrospher 100 RP 18 column. The mobile phases used were solvent A [H2O/HCOOH (996:4)] and solvent B [MeOH/HCOOH (996:4)] and gradient elution of 0-3% of B in 5 min, 3-2 20% of B in 20 min, and 20-100% of B in 50 min with a flow rate set at 1 mL/min and a detection wavelength set at 280 nm.
7. The authors did not specify if the results presented in Figures 2 through 4 are mean values of replicates (n=?). Error bars are SD or SEM? Also, the statistical significance of the differences was ever considered? The way data was presented made it impossible to distinguish error bars for different compounds. This, associated with the lack of statistical analysis of the data makes the conclusions meaningless.
The error bars represent standard deviation from a group mean value for each compound. In mayority each group have n=20 different wines.
8. English is fairly poor and sloppy in some sentences, especially when presenting results. Some examples are given, but they are by no means a complete list and a thorough revision of English and grammar is mandatory. For example: Line 143: “…but without regarding the period stayed in bottle…” or line 130: “…Comparing all the results together, the concentration average for the sum on ellagitannins…” or line 157: “…Figures 4 shown the content of each ellagitannin…” or line 166: “…This practice could increment the content on ellagitannins that can end in the stabilization of the wine inside the bottle…” or lines 187 through 189: “…Moreover, the practice in Rioja region for the origin appellation that allows the addition of 15% of another type of wine, give concentration in ellagitannins ans flavano-188 ellagitannis more stable without depending on the oak barrel or bottle ageing periods.”
English was performed by the authors.
Round 2
Reviewer 1 Report
the manuscript can be now published
Author Response
We thanks Reviewer 1 indicating that our paper can be publish.
Reviewer 2 Report
The paper is now OK.
Only two minor comments:
Lines 194-197: I think there is a confusion between Libourne and Medoc
Figure 4: the image 4c appears superimposed on the 4b
Author Response
Corrections asked by Reviewer 2 have been done.

Reviewer 4 Report
1. Abstract: The abstract is poorly presented and very little informative of the main results obtained in the study. It should include the main results and a short conclusion of the work as a whole. Abstract was modified by the authors. Comment by Referee: OK. ...”After castalagin the mongolicain A is the second molecule present in majority for Bordeaux wines with a contribution of 18.46%, 3 folds higher than the value obtained for Rioja wines, suggesting that this compound could be a marker of the time of ageing in oak barrel. The degradation of C-ellagitannins and flavano-ellagitannins is more accused in Rioja wines, which were kept longer in oak barrels and showed the lowest concentrations on this type of compounds” 2. Species names (eg. Line 40, line 77) should be in italics. Done Comment by referee: OK 3. Material and Methods: the authors report that castalagin was extracted from Quercus robus Heartwood and implied this compound as the starting material for the hemisynthesis of acutissimin A/B, epiacutissimin A/B, and ß-1-O-78 ethylvescalagin. However, in the paper by Quideau et al. (Chem. Eur. J. 2005, 11, 6503–6513), all nucleophilic reactions to obtain acutissimins A and B as well as epiacutissimins A/B were done with (-)-vescalagin as the strating material. What reactions conditions were used and what starting material was actually used for the hemisynthesis of the compounds cited in the manuscript? Answer : We did not said that castalagine was used for flavano-ellagitannins hemysinthese. Castalagine was one of the ellagitannins analyzed in our work and to have the standard this compound was extracted from oak heartwood. It is (-)-vescalagine the compound used for the hemysintheses. The reactions conditions were the same as in references, because they were carried out by S. Quideau laboratory. Comment by referee: Please include a statement making it clear that it was (-)-vescalagin the starting material for the hemisynthesis. For example, we suggest this: "Castalagin was extracted from Quercus robur heartwood [27] and the semisynthesis of the acutissimin A/B (9/10, >95% pure), the epiacutissimin A/B (11/12, >95% pure), and the ß-1-O-ethylvescalagin (13, 98% pure) were performed in acidic media [1.5% (v/v) TFA/THF] at 60 °C using (-)-vescalagin as starting material as previously described." 4. The reaction conditions for the oxidation of acutissimin A into mongolicain A should be given. The oxidation of acutissimin A to achieve the compound called mongolicain A was performed in acid aqueous solution. 100 mg of acutissimin A were placed in 2 ml of water acidified a pH 3.5 with tartaric acid. This aqueous solution was behind air contact and environmental temperature (25°C) during 15 days. Comments from referee: Please add oxidation conditions to the text. 5. The purity of the obtained compounds is given. What method was used to ascertain the purity of the compounds? The percentage purity given is HPLC peak purity? Is there any high-resolution mass spectral data to support the purity of compounds? This should be provided. A condensed form of the NMR spectral data should be provided to confirm identity of the substances. All the hemysintheses were carried out inside S. Quideau laboratory with the same protocols as they were described on the references of the article. All the information reviewer is asking for is given in S.Quideau’s references. Referee Comments: The fact that one of the authors of the paper was involved in the synthesis of marker compounds following a previoulsy published method is no excuse to fail to provide evidence of the purity of compounds. The methods used to assess purity (NMR, HRMS, melting point, etc) should be mentioned. 6. The authors report that the quantification of compounds was done using a previously described LC/MS method (reference 2). However, no experimental details were given of their use of this method: For example: What were the separation conditions? What were the ionization conditions? Was a linear quadrupole ion trap mass spectrometer (Sciex, API4000 QTRap) used as in the original reference? What detection mode was used? The authors should provide typical ion chromatograms of the quantitated compounds in the wine fractions to prove that adequate selectivity was achieved. As reviewer suggested, authors have added some missing details of quantifications : The analysis of ellagitannins composition in red wines was performed using a Thermo-Finnigan Surveyor HPLC system containing an autosampler (Surveyor autosampler Plus), a quaternary pump (Surveyor LC pump Plus), and a UV-vis detector (Surveyor PDA Plus) controlled by Xcalibur data treatment system. This HPLC system was also coupled to a Thermo-Finnigan LCQ Advantage spectrometer equipped with an ion trap mass analyzer. The electrospray ionization and mass detection was performed in negative ion mode with the following optimized parameters: capillary temperature, 300 °C; capillary voltage, -46 V; nebulizer gas flow, 1 L/min; desolvation gas flow, 0.25 L/min; and spray voltage, 5 kV. These analyses were carried out in technical duplicates on a 250 × 4.6 mm, 5 µm, Lichrospher 100 RP 18 column. The mobile phases used were solvent A [H2O/HCOOH (996:4)] and solvent B [MeOH/HCOOH (996:4)] and gradient elution of 0-3% of B in 5 min, 3-2 20% of B in 20 min, and 20-100% of B in 50 min with a flow rate set at 1 mL/min and a detection wavelength set at 280 nm. Referee comments: OK 7. The authors did not specify if the results presented in Figures 2 through 4 are mean values of replicates (n=?). Error bars are SD or SEM? Also, the statistical significance of the differences was ever considered? The way data was presented made it impossible to distinguish error bars for different compounds. This, associated with the lack of statistical analysis of the data makes the conclusions meaningless. The error bars represent standard deviation from a group mean value for each compound. In mayority each group have n=20 different wines. Referee comments: No statistical data analysis was done. Without proper statistical analysis, the significance of the differences observed are meaningless. We urge authors to conduct a proper statistical analysis of their results. 8. English is fairly poor and sloppy in some sentences, especially when presenting results. Some examples are given, but they are by no means a complete list and a thorough revision of English and grammar is mandatory. For example: Line 143: “…but without regarding the period stayed in bottle…” or line 130: “…Comparing all the results together, the concentration average for the sum on ellagitannins…” or line 157: “…Figures 4 shown the content of each ellagitannin…” or line 166: “…This practice could increment the content on ellagitannins that can end in the stabilization of the wine inside the bottle…” or lines 187 through 189: “…Moreover, the practice in Rioja region for the origin appellation that allows the addition of 15% of another type of wine, give concentration in ellagitannins ans flavano-188 ellagitannis more stable without depending on the oak barrel or bottle ageing periods.” English was performed by the authors. Referee comments: OK
Author Response
Corrections asked by reviewer 4 have been done.
